# Relationship between Accurate Diagnosis of Sarcopenia and Prognosis in Patients with Hepatocellular Carcinoma Treated with Atezolizumab plus Bevacizumab Combination Therapy

**DOI:** 10.3390/cancers15123243

**Published:** 2023-06-19

**Authors:** Kyoko Oura, Asahiro Morishita, Takushi Manabe, Kei Takuma, Mai Nakahara, Tomoko Tadokoro, Koji Fujita, Shima Mimura, Joji Tani, Masafumi Ono, Chikara Ogawa, Akio Moriya, Tomonori Senoo, Akemi Tsutsui, Takuya Nagano, Koichi Takaguchi, Takashi Himoto, Tsutomu Masaki

**Affiliations:** 1Department of Gastroenterology and Neurology, Faculty of Medicine, Kagawa University, Kita-gun 761-0793, Kagawa, Japan; 2Department of Gastroenterology and Hepatology, Takamatsu Red Cross Hospital, Takamatsu 760-0017, Kagawa, Japan; 3Department of Gastroenterology, Mitoyo General Hospital, Kanonji 769-1695, Kagawa, Japan; 4Department of Hepatology, Kagawa Prefectural Central Hospital, Takamatsu 760-8557, Kagawa, Japan; 5Department of Medical Technology, Kagawa Prefectural University of Health Sciences, Takamatsu 761-0123, Kagawa, Japan

**Keywords:** hepatocellular carcinoma, sarcopenia, atezolizumab, bevacizumab, immunotherapy, grip strength, skeletal muscle, psoas muscle, clinical outcome, overall survival

## Abstract

**Simple Summary:**

Although sarcopenia-related factors, including decreased skeletal muscle index (SMI), have been reported to affect therapeutic efficacy and the occurrence of adverse events in sorafenib or lenvatinib treatment for hepatocellular carcinoma (HCC), there is considerably less evidence regarding the relationship between SMI and prognosis in the HCC patients treated with atezolizumab plus bevacizumab (atezo/bev) therapy. Furthermore, there are no reports of muscle strength, including grip strength (GS), which is essential for an accurate diagnosis of sarcopenia. This is the first study to show the relationship between sarcopenia, diagnosed by decreased GS and SMI, and clinical outcomes in atezo/bev therapy, with novel evidence having a strong impact. The presence of sarcopenia is significantly associated with shorter overall survival with the occurrence of adverse events and decreased liver function. Monitoring of both GS and SMI is useful for assessing the general condition and predicting prognosis in HCC patients treated with atezo/bev therapy.

**Abstract:**

Although there have been advances in the prevention and diagnosis of hepatocellular carcinoma (HCC) in recent years, many HCC patients are still diagnosed with advanced stage. Systemic therapy is indicated for unresectable HCC (uHCC) with major vascular invasion and/or extrahepatic metastases, and the atezolizumab plus bevacizumab (atezo/bev) combination is currently recommended as first-line treatment for uHCC. Recently, sarcopenia-related factors, including decreased skeletal muscle index (SMI), have been reportedly associated with prognosis in uHCC patients treated with sorafenib or lenvatinib. There are few reports on muscle strength assessments, including grip strength (GS), despite their importance in accurate sarcopenia diagnosis, and furthermore, there is no evidence regarding atezo/bev therapy. In this study, we investigated whether sarcopenia affects the clinical outcome of atezo/bev therapy. This study included 64 uHCC patients on atezo/bev therapy and assessed their GS and SMI, and SMI was measured using bioelectrical impedance analysis (BIA). We diagnosed sarcopenia based on GS and BIA-SMI and compared the clinical outcomes in the sarcopenia and non-sarcopenia groups. Of these patients, 28 had sarcopenia, and 36 had non-sarcopenia. Adverse events (AEs) frequently occurred, and the albumin-bilirubin score significantly decreased after atezo/bev therapy in the sarcopenia group than in the non-sarcopenia group. The median progression-free survival was 4.7 (0.4–26.4) and 10.6 (1.1–24.5) months in the sarcopenia and non-sarcopenia groups, respectively. The median overall survival (OS) was 12.6 (1.4–27.7) months in the sarcopenia group and was not reached in the non-sarcopenia group, indicating a significant difference in the Kaplan-Meier survival curves for both groups (*p* < 0.01). In multivariate analysis, sarcopenia was significantly associated with OS. In conclusion, sarcopenia was significantly associated with poor clinical outcomes based on the occurrence of AEs and decreased liver function in uHCC patients on atezo/bev therapy. GS and SMI are important parameters for accurately diagnosing sarcopenia.

## 1. Introduction

Recent global statistics show that liver cancer is the sixth most common cancer and the third most common cause of cancer-related deaths [1]. Hepatocellular carcinoma (HCC) is the most common histological type, accounting for approximately 90% of cases. While persistent hepatitis B and C virus infections are major risk factors for HCC, the number of patients with HCC due to non-alcoholic steatohepatitis associated with metabolic syndromes such as diabetes, hypertension, dyslipidemia, and obesity is rapidly increasing in Western countries and Japan [2]. Despite recent significant advances in the prevention and diagnosis of HCC, more than 50% of patients with HCC are still classified as Barcelona Clinic Liver Cancer (BCLC) staging B or C at the time of diagnosis. The difficulty in early detection is one reason HCC is associated with poor prognosis. Patients with unresectable HCC (uHCC) with major vascular invasion or extrahepatic metastasis are classified as BCLC stage C, and systemic therapy is recommended [3]. Furthermore, even for BCLC stage B, systemic therapy is increasingly recommended for patients with intrahepatic tumors exceeding up-to-7 criteria with limited response to conventional transcatheter arterial chemoembolization [4]. Based on phase III clinical trials, several regimens of sorafenib [5], regorafenib [6], lenvatinib [7], cabozantinib [8], ramucirumab [9], and atezolizumab plus bevacizumab (atezo/bev) [10] are currently approved for uHCC. Aezo/Bev therapy is a combined immunotherapy, which combines an anti-programmed cell death 1-ligand 1 antibody with an anti-vascular endothelial growth factor antibody [10]. The IMbrave 150 trial in patients with uHCC who had not previously received systemic therapy showed statistically significant differences in overall survival (OS) between sorafenib, the previous standard of treatment, and atezo/bev therapy [10]. Based on the results of the IMbrave 150 trial, Atezo/Bev therapy is considered the most effective first-line treatment strategy.

Sarcopenia is characterized by loss of muscle strength and decreased muscle mass and occurs secondary to various underlying diseases such as liver, renal and inflammatory diseases and malignancies [11]. The factors that cause sarcopenia in patients with liver disease include protein-energy malnutrition, inadequate protein synthesis due to branched-chain amino acid (BCAA) deficiency, decreased testosterone, increased myostatin expression, and increased reactive oxygen species and inflammatory cytokines, all of which are intricately linked to the development of sarcopenia [12]. For systemic therapy in uHCC, sarcopenia-related factors have been reportedly associated with the clinical outcomes of sorafenib [13,14,15,16] and lenvatinib treatment [17,18,19], mostly relying on muscle mass assessment such as skeletal muscle index (SMI) at the level of the third lumbar vertebrae (L3) in computed tomography (CT) images (CT-L3 SMI). Although recent diagnostic criteria have emphasized that muscle strength should be assessed before muscle mass as sarcopenia-related parameters [20,21,22], there is insufficient evidence for muscle strength assessment in uHCC patients on systemic therapy, with only one report on the association between grip strength (GS) and prognosis in uHCC patients receiving lenvatinib treatment [23]. Although European and Asian guidelines have historically recommended SMI using bioelectrical impedance analysis (BIA) for the subsequent assessment of skeletal muscle mass when sarcopenia is suspected due to loss of muscle strength [20,21], there are no studies on BIA-SMI and clinical outcomes in uHCC patients on systemic therapy. In addition, there has been a recent report that changes in CT-L3 SMI before and after atezo/bev therapy are associated with prognosis in HCC patients [24]. However, an accurate diagnosis of sarcopenia based on muscle strength, including GS as well as other skeletal muscle assessments, may better reflect the general condition of HCC patients treated with atezo/bev therapy.

To address these issues, we examined the prognostic value of sarcopenia diagnosed by GS and BIA-SMI. This is the first study to show the relationship between sarcopenia, diagnosed by decreased GS and SMI, and clinical outcomes in atezo/bev therapy. We also examined whether sarcopenia affects changes in muscle mass, liver function, and occurrence of adverse events (AEs) after atezo/bev therapy.

## 2. Materials and Methods

### 2.1. Study Design and Protocol

This retrospective study included patients with uHCC who received atezo/bev therapy from October 2020 to January 2023 at multiple institutions, including Kagawa University, Takamatsu Red Cross Hospital, Mitoyo General Hospital, and Kagawa Prefectural Central Hospital. Their GS and BIA-SMI were assessed before commencing treatment. Patients who received less than two courses of atezo/bev therapy or those who were ineligible for body composition evaluation by BIA due to ascites, severe edema, or difficulty standing were excluded. HCC was diagnosed using complementary tumor markers, contrast-enhanced CT, and magnetic resonance imaging (MRI). When typical HCC findings were not seen, a needle biopsy was performed to confirm the diagnosis. Sarcopenia was diagnosed based on the guidelines of the Asian Working Group for Sarcopenia (AWGS) [21] and the Japan Society of Hepatology (JSH) for sarcopenia in liver disease [22], and the GS and BIA-SMI of the patients were measured. GS was measured twice with both the left and right hands, and the average value of the better scores was taken as the GS numerical value, with a cutoff value of 28 kg for males and 18 kg for females. The BIA device used was an Inbody 770 (InBody Co., Ltd., Seoul, Republic of Korea), and the patient stood on two metal electrodes and held a metal grip electrode to assess body composition. BIA-SMI was calculated by dividing the skeletal muscle mass of BIA by the square of height, and the cutoff value was 7.0 kg/m^2^ for males and 5.7 kg/m^2^ for females. Sarcopenia was defined as both GS and BIA-SMI values below these cutoff values. Patient characteristics, changes in liver function and skeletal muscle mass during treatment, therapeutic effects, the occurrence of AEs, and prognosis were compared between sarcopenia and non-sarcopenia groups.

Treatment consisted of 1200 mg atezo plus 15 mg/kg bev administered intravenously every 3 weeks until disease progression or intolerable adverse events occurred. Treatment was interrupted according to the manufacturer’s guidelines. General condition was assessed using the body mass index (BMI) and Eastern Cooperative Oncology Group Performance Status (PS). Nutritional status was evaluated using the Controlling Nutritional Status (CONUT) score, and liver function was evaluated using the Child-Pugh score, albumin-bilirubin (ALBI) score, and modified ALBI (mALBI) grade [25]. Clinical staging was evaluated by tumor-lymph node-metastasis classification based on the criteria by the Liver Cancer Study Group of Japan according to tumor diameter, number of tumors, vascular invasion, lymph node metastasis, and distant metastasis. Contrast-enhanced CT was performed after 6–9 weeks of treatment. The muscle area on CT was evaluated by measuring the psoas and skeletal muscle areas at the L3 level using CT images with SYNAPSE VINCENT (FUJIFILM, Tokyo, Japan). The change in skeletal muscle mass after treatment was evaluated by comparing the results of these areas divided by the square of the height, that is, CT-L3 psoas muscle index (PMI) and CT-L3 SMI, at baseline and after 6 weeks of treatment. Therapeutic effects were classified as complete response (CR), partial response (PR), stable disease (SD), and progressive disease (PD) according to the modified Response Evaluation Criteria in Solid Tumors (mRESIST). AEs were evaluated by the attending physician every 3 weeks based on the Common Terminology Criteria for Adverse Events version 5.0 [26].

### 2.2. Statical Analysis

Statistical analyses were performed using GraphPad Prism (Prism 8.4.3; San Diego, CA, USA). The frequencies were compared using Chi-square or Fisher’s exact tests. Continuous variables are presented as the median, and the differences in the median were compared using the Mann–Whitney U test. The paired groups before and after treatment were compared using the Wilcoxon signed-rank sum test. Progression-free survival (PFS) and OS rates in January 2023 were calculated using the Kaplan–Meier method, and significance was determined using the log-rank test. Multivariate analyses were performed for factors related to OS using the Cox proportional hazards model, and significance was determined for each factor using the Wald test. Statistical significance was set at *p* < 0.05.

### 2.3. Ethical Approval

The present study was approved by the ethical committee of Kagawa University, Faculty of Medicine (Ethics approval 2022-147). This study complied with the guidelines for human studies and was conducted ethically in accordance with the World Medical Association Declaration of Helsinki.

## 3. Results

### 3.1. Patient Characteristics

In this study, 81 patients with uHCC who received atezo/bev therapy were assessed using GS and BIA-SMI to ascertain the presence of sarcopenia. A flowchart of the patient selection process is shown in Figure 1. Seventeen patients were excluded based on the following exclusion criteria: one patient had difficulty standing for a while and was ineligible for accurate skeletal muscle mass measurement using BIA; one patient was diagnosed before treatment by MRI alone and had no skeletal muscle mass assessed by CT; 12 patients had completed less than two courses of atezo/bev therapy and were ineligible for evaluation of therapeutic effects; and three patients did not have a CT scan performed at 6–9 weeks after commencing treatment. A total of 64 patients were enrolled and analyzed. Patients whose GS and BIA-SMI were below the cutoff values were diagnosed with sarcopenia. As such, there were 28 patients in the sarcopenia group and 36 in the non-sarcopenia group for comparison.

Baseline patient characteristics are shown in Table 1. The median ages were 79.5 (67–91) and 71.5 (42–86) years in the sarcopenia and non-sarcopenia groups, respectively, with the sarcopenia group being significantly older. Weight and BMI were significantly lower in the sarcopenia group than in the non-sarcopenia group. Median GS and BIA-SMI were significantly higher in the non-sarcopenia group (*p* < 0.01) The percentage of patients with PS ≥ 2 was 35.7% and 5.6% in the sarcopenia and non-sarcopenia groups, respectively, with significantly more patients in the sarcopenia group having restrictions in daily living. Otherwise, the two groups had no significant differences in baseline characteristics, including liver function and clinical stages.

### 3.2. Skeletal Muscle Mass and Related Indicators

The correlation between pretreatment BIA-SMI and serum albumin, total cholesterol, peripheral blood lymphocyte count, ALBI score, GS, and CT-L3 PMI is shown in Figure 2. In the sarcopenia group, serum albumin and blood lymphocyte count correlated positively with BIA-SMI, (shown in Figure 2A,C), whereas total cholesterol and BIA-SMI were not correlated (shown in Figure 2B). ALBI scores were negatively correlated with BIA-SMI in the sarcopenia group (shown in Figure 2D), suggesting that good liver function was associated with high muscle mass. Furthermore, BIA-SMI and GS were positively correlated in the sarcopenia group (shown in Figure 2E), and BIA-SMI and CT-L3 PMI were positively correlated in patients with and without sarcopenia (shown in Figure 2F). The percentage of patients with a CONUT score of ≥2 was 67.9% and 72.2% in the sarcopenia and non-sarcopenia groups, respectively, with no significant difference. The percentages of patients with mALBI grades 1, 2a, 2b, and 3 were 28.6%, 32.1%, 35.7%, and 3.6% in the sarcopenia group and 31.7%, 21.2%, 36.1%, and 2.8% in the non-sarcopenia group, respectively, with no significant difference between the two groups.

The results of the body composition assessment are shown in Figure 3. Values of body fat mass, soft lean mass, and skeletal muscle mass were significantly lower in the sarcopenia group (Figure 3A–C). Muscle mass in all regions including arm muscle mass, trunk muscle mass, and leg muscle mass was significantly lower in the sarcopenia group (Figure 3D–F).

### 3.3. Therapeutic Effects

Table 2 shows the therapeutic effects of mRESIST based on contrast-enhanced CT findings at 6–10 weeks after commencing atezo/bev therapy. The overall response rate (CR + PR) was 25.0% and 38.9% in the sarcopenia and non-sarcopenia groups, respectively. The disease control rate (CR + PR + SD) was 75.0% and 80.6% in the sarcopenia and non-sarcopenia groups, respectively. The effect of early treatment was not significantly different between the two groups.

### 3.4. AEs in uHCC Patients with Atezo/Bev Therapy

Of the 28 patients in the sarcopenia group, 25 (89.3%) had AEs of any grade, while 24 (66.7%) of the 36 patients in the non-sarcopenia group had AEs of any grade, indicating that the sarcopenia group had significantly more AEs than the non-sarcopenia group (shown in Table 3). In the sarcopenia group, anorexia was observed in 18 (64.3%) patients, proteinuria in 9 (32.1%), diarrhea in 8 (28.6%), mucositis in 5 (17.9%), eczema in 3 (10.7%), and nausea in 3 (10.7%). In the non-sarcopenia group, proteinuria was observed in 10 patients (27.8%), anorexia in 7 (19.4%), hypertension in 7 (19.4%), and oral mucositis in 4 (11.1%). Grade 3 or higher AEs were observed in 14 (50.0%) patients in the sarcopenia group and 7 (19.4%) patients in the non-sarcopenia groups, respectively, indicating that the sarcopenia group had significantly more severe AEs than the non-sarcopenia group.

### 3.5. Changes in ALBI Score and Muscle Mass with Atezo/Bev Therapy

Changes in liver function and nutritional status before and after atezo/bev therapy are shown in Figure 4. In the sarcopenia group, the median ALBI score significantly increased, with −2.393 (−3.330–−1.365) at baseline and −2.171 (−2.990–−1.114) after 6 weeks of treatment (*p* < 0.01), indicating that liver function worsened with atezo/bev therapy. In contrast, in the non-sarcopenia group, the median ALBI score was −2.435 (−3.189–−1.251) at baseline and −2.134 (−3.328–−1.406) after 6 weeks of treatment, with no significant change (shown in Figure 4A). The median CONUT scores at baseline and after 6 weeks of treatment were 3.5 (0–9) and 4 (0–11) in the sarcopenia group and 2 (0–8) and 3 (0–7) in the non-sarcopenia group, respectively, with no significant difference between the two groups (shown in Figure 4B).

Changes in fat and muscle areas at the third lumbar vertebrae at baseline and 6 weeks after commencing atezo/bev therapy are shown in Figure 5. The median subcutaneous fat area, internal fat area, or psoas muscle area decreased significantly in the sarcopenia group, while there was no significant difference between pre-and post-treatment in the non-sarcopenia group (shown in Figure 5A–C). There was no significant pre- and post-treatment change in the median skeletal muscle area in the sarcopenia group, while it significantly decreased after 6 weeks of treatment compared to baseline in the non-sarcopenia group (shown in Figure 5D).

### 3.6. Prognosis Analysis

Clinical outcomes at the end of the data cutoff period (April 2023) are shown in Figure 6. The median duration of observation was 11.5 months. The median PFS was 4.7 (0.4–26.4) and 10.6 (1.1–24.5) months in the sarcopenia and non-sarcopenia groups, respectively (shown in Figure 6A). The median OS was 12.6 (1.4–27.7) months in the sarcopenia group. OS was not reached in the non-sarcopenia group as the patients were still living (shown in Figure 6B). The Kaplan–Meier survival curves for both groups showed a significant difference, with a hazard ratio (HR) of 2.869 (95% confidence interval (CI): 1.287–6.396) (*p* < 0.01).

The association between age and OS was analyzed by subgroups, divided into elderly (≥80 years) and non-elderly (≤79 years) in the sarcopenia group. The median OS was 264 (43–657) days for the elderly and 332 (42–861) days for the non-elderly, with no significant difference. In addition, the association between PS and clinical outcome was zanalyzed separately for subgroups of patients with PS 0 and those with PS ≥ 1. In the sarcopenia group, the median OS was 367 (51–796) days for the patients with PS 0 and 237 (42–861) days for the patients with PS ≥ 1, with no significant difference.

We analyzed the factors associated with OS with atezo/bev therapy. The results are shown in Table 4. Multivariate analyses showed that in addition to the Child–Pugh score, sarcopenia was a significant factor associated with OS (yes vs. no: HR 2.58; 95% CI, 1.17–5.95; *p* < 0.05), but age and PS were not extracted as significant factors.

## 4. Discussion

Since the IMbrave150 trial in uHCC patients who had not received systemic therapy revealed the superiority of atezo/bev therapy over sorafenib, atezo/bev has been the most effective first-line treatment strategy for uHCC [10]. To the best of our knowledge, this is the first study to show that sarcopenia, diagnosed by both GS and BIA-SMI, is associated with the prognosis of patients with uHCC treated with atezo/bev therapy.

According to the guidelines of the European Working Group on Sarcopenia in Older People (EWGSOP) or AWGS, the flowchart for the diagnosis of sarcopenia begins by selecting patients with possible sarcopenia and assessing muscle strength using GS [20,21]. When decreased muscle strength is present, EWGSOP guidelines require dual-energy X-ray absorptiometry (DXA), BIA, CT, or MRI to assess skeletal muscle mass and confirm the diagnosis of sarcopenia. In the AWGS guidelines, only DXA or BIA is recommended for skeletal muscle mass assessment. The JSH guideline algorithm includes only GS as a measure of muscle strength and CT-L3 SMI or CT-L3 PMI as a measure of skeletal muscle mass, in addition to BIA-SMI. Therefore, in this study, we used GS and BIA-SMI measurements to diagnose sarcopenia based on AWGH and JSH guidelines, because BIA-SMI has historically been recommended by the EWGSOP and AWGS guidelines for accurate muscle mass assessment, and because BIA can assess muscle mass and fat mass by site. However, as the CT-L3 PMI correlated with baseline BIA-SMI before atezo/bev therapy, the CT-L3 PMI may be used to easily diagnose muscle mass decline and evaluate its association with clinical outcomes of patients with uHCC on atezo/bev therapy.

Several studies on the clinical prognosis of HCC patients receiving sorafenib treatment have reported an association with decreased skeletal muscle mass. Decreased CT-L3 SMI before sorafenib treatment was associated with poor clinical outcomes, including therapeutic effects, PFS, and OS, and could be an intervention target to improve clinical outcomes [14,15,16]. In a study focusing on changes in CT-L3 SMI before and after sorafenib treatment, patients with HCC with decreased CT-L3 SMI after sorafenib treatment had significantly shorter survival, suggesting that rapid skeletal muscle mass loss was associated with poor prognosis [13]. Thus, although clinical outcomes of sorafenib treatment have been reported to be related to muscle mass based on CT before or after treatment, none have been reported in combination with muscle strength assessment including GS.

In patients with uHCC receiving lenvatinib treatment, several retrospective studies have reported the relationship between clinical outcomes and CT-based skeletal muscle mass assessment. In studies investigating whether skeletal muscle mass correlates with tolerability and prognosis in lenvatinib-treated HCC patients, patients with low pretreatment CT-L3 SMI had more serious AEs leading to on-treatment failure and significantly worse OS than those with high pretreatment CT-L3 SMI [17], indicating that decreased CT-L3 SMI was a significant prognostic factor in lenvatinib treatment [18,19]. As with sorafenib, in lenvatinib treatment in patients with uHCC, most studies on prognosis and sarcopenia-related factors are associated with CT-L3 SMI because CT scans are often performed in HCC patients to evaluate therapeutic effects, and it is easy to evaluate skeletal muscle area retrospectively with CT findings. Although very few studies have evaluated the effect of muscle strength in HCC patients receiving lenvatinib treatment, one study evaluated GS. Endo et al. investigated the prognosis of 63 lenvatinib-treated HCC patients, assessed by GS and CT-L3 SMI, and found no significant difference in OS between the groups with decreased and normal CT-L3 SMI; however, the group with decreased GS had significantly worse OS than the group with normal GS [23]. In the multivariate Cox proportional hazards model, decreased GS was independent of poor prognostic factors.

There is a lack of evidence for the association between clinical outcomes and sarcopenia-related factors in immunotherapy for uHCC compared to tyrosine kinase inhibitors, such as sorafenib or lenvatinib. For atezo/bev therapy, the clinical use of which has been rapidly spreading for uHCC, only one paper from Japan reported the association between CT-L3 SMI and clinical prognosis. Matsumoto et al. evaluated CT-L3 SMI before and 6–14 weeks after treatment in 32 HCC patients treated with atezo/bev therapy. They revealed that patients with decreased CT-L3 SMI after treatment had significantly shorter PFS than those with non-decreased CT-L3 SMI, indicating the importance of monitoring skeletal muscle mass during atezo/bev therapy [24]. There have been no reports on muscle strength assessment, such as GS or muscle mass assessment, using BIA-SMI, which is essential for accurately diagnosing sarcopenia. Therefore, we demonstrated the importance of evaluating GS and BIA-SMA and predicting clinical outcomes in the present multicenter study. In this study, patients with sarcopenia had significantly shorter PFS and OS than non-sarcopenia patients. Furthermore, the present study did not find a decrease in CT-L3 SMI after 6 weeks of atezo/bev therapy in patients with sarcopenia, which differs from the results of a previous report [24]. However, their report evaluated the change in CT-L3 SMI over a wide range of time (6–14 weeks), which may explain why their analysis of long-term SMI change was different from our result. Interestingly, our study showed a significant increase in AEs occurrence and a decrease in ALBI score after 6 weeks of atezo/bev therapy in patients with sarcopenia, suggesting that AEs occurrence and poor liver function may be associated with poor clinical outcomes. Therefore, when treating uHCC patients with sarcopenia, strategies such as careful management of AEs and drug therapy to preserve liver function may contribute to the improvement of clinical outcomes. Specifically, uHCC patients on atezo/bev therapy should be observed more closely over a period shorter than 3 weeks, and AE should be carefully addressed. In addition, branched-chain amino acid (BCAA) formulation should be actively considered, as evidence has been established that prescribing BCAA contributes to the maintenance of liver function [27,28]. In fact, BCAA administration has been reported to be important not only for improving liver function but also for maintaining skeletal muscle mass [29], which may have a dual benefit for patients with sarcopenia. Another report suggests that L-carnitine administration is useful in preventing skeletal muscle loss in patients with cirrhosis [30], and may be an important strategy in the multidisciplinary treatment of uHCC. Furthermore, a previous systematic review suggested that ICIs caused more AEs and affected clinical outcomes in patients with a history of autoimmune disease [31], and sarcopenic patients should be evaluated more carefully for preexisting conditions, including autoimmune disease.

Our study had some limitations. First, it was a retrospective study, and the number of patients included was insufficient because GS measurement and body composition examination by BIA were often not routinely performed before atezo/bev treatment. Therefore, a study with a larger number of patients is necessary. Second, the sarcopenia group included more elderly patients. This reflected the fact that the prevalence of sarcopenia increases with age, making age-matched comparisons between the sarcopenia and non-sarcopenia groups difficult. Third, because the number of patients receiving BCAA and carnitine supplementation was extremely small, we could not analyze the effects of BCAA and levocarnitine supplementation on sarcopenia suppression. Fourth, the median OS was not achieved in the non-sarcopenia group, and further analysis of the long-term prognosis is necessary. Despite these limitations, this study is the first to show that the presence of sarcopenia diagnosed by GS and BIA-SMI is associated with prognosis in HCC patients on atezo/bev therapy and can provide novel evidence with strong clinical impact.

## 5. Conclusions

Sarcopenia was significantly associated with poor clinical outcomes, including OS, based on the increased occurrence of AEs and decreased liver function in patients with uHCC treated with atezo/bev therapy. It is important to accurately diagnose sarcopenia in these patients by measuring GS and SMI before initiating treatment.

## Figures and Tables

**Figure 1 cancers-15-03243-f001:**
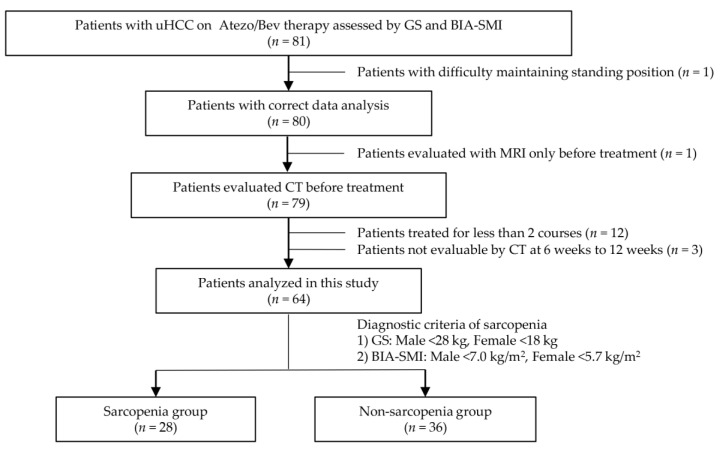
Flowchart of patient selection. Initially, there were 81 eligible patients. Seventeen ineligible patients were excluded because of difficulties in accurate assessment by BIA, lack of necessary testing at the appropriate time, or less than 2 courses of treatment. Finally, 64 patients were enrolled in this study. The patients whose GS and BIA-SMI were below the cutoff values were diagnosed with sarcopenia. The comparison was made between the patients in the sarcopenia group and the patients in the non-sarcopenia group. uHCC: unresectable hepatocellular carcinoma, Atezo: atezolizumab, Bev: Bevacizumab, GS: grip strength, BIA: bioelectrical impedance analysis, SMI: skeletal muscle index, MRI: magnetic resonance imaging, CT: computed tomography.

**Figure 2 cancers-15-03243-f002:**
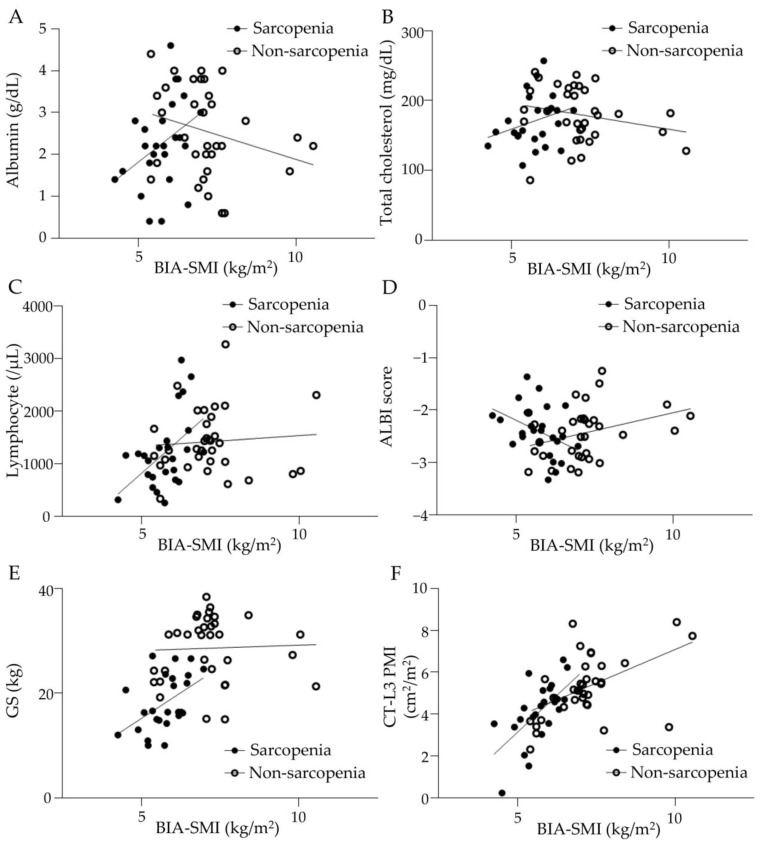
Correlation between BIA-SMI and various parameters in sarcopenia and non-sarcopenia groups. (**A**) Serum albumin level correlated with BIA-SMI in the sarcopenia group but not in the non-sarcopenia group. Sarcopenia group: y = 0.3205x + 1.755, r = 0.1514, *p* < 0.05. Non-sarcopenia group: y = −0.1301x + 4.721, r = 0.08191, *p* = 0.0906. (**B**) Serum total cholesterol level did not correlate with BIA-SMI. Sarcopenia group: y = 15.86x + 81.17, r = 0.07723, *p* = 0.1522. Non-sarcopenia group: y = −7.926x + 235.3, r = 0.05355, *p* = 0.1745. (**C**) Blood lymphocyte counts correlated with BIA-SMI in the sarcopenia group but not in the non-sarcopenia group. Sarcopenia group: y = 514.4x − 1730, r = 0.2332, *p* < 0.01. Non-sarcopenia group: y = 42.32x + 1035, r = 0.006897, *p* = 0.6301. (**D**) ALBI score inversely correlated with BIA-SMI in the sarcopenia group. Sarcopenia group: y = − 0.3130x − 0.580, r = 0.1748, *p* < 0.05. Non-sarcopenia group: y = 0.1473x − 3.509, r = 0.1218, *p* < 0.05. (**E**) Grip strength correlated with BIA-SMI in the sarcopenia group but not in the non-sarcopenia group. Sarcopenia group: y = 4.239x − 6.112, r = 0.2262, *p* < 0.05. Non-sarcopenia group: y = 0.1114x + 28.27, r = 0.0004627, *p* = 0.9009. (**F**) CT-PMI correlated with BIA-SMI in both groups. Sarcopenia group: y = 1.491x − 4.475, r = 0.4138, *p* < 0.01. Non-sarcopenia group: y = 0.6303x + 0.7716, r = 0.2429, *p* < 0.01. BIA: bioelectrical impedance analysis, SMI: skeletal muscle index, ALBI: albumin-bilirubin, GS: grip strength, CT: computed tomography, L3: third lumber vertebra, PMI: psoas muscle index.

**Figure 3 cancers-15-03243-f003:**
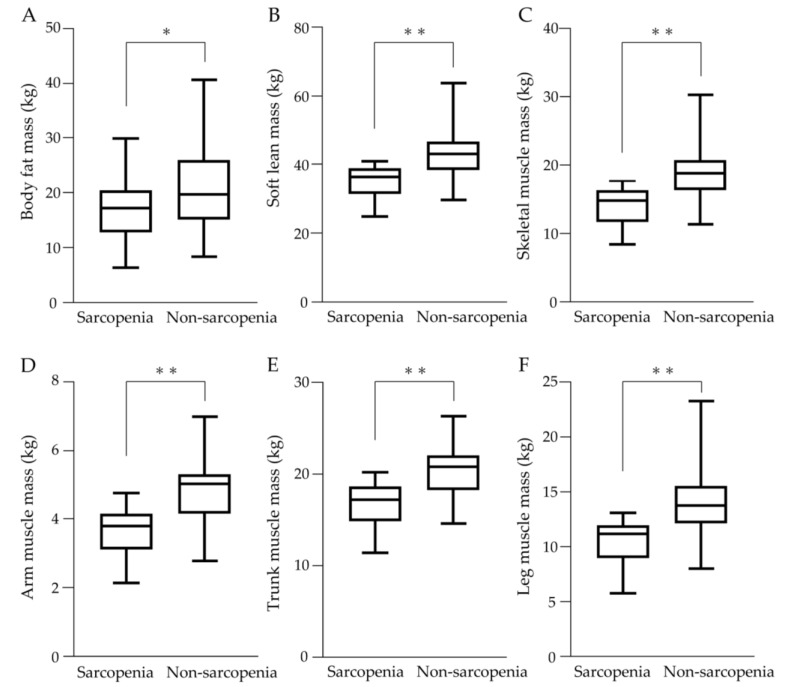
Body composition assessment in sarcopenia and non-sarcopenia groups. (**A**) Body fat mass, (**B**) Soft lean mass, (**C**) Skeletal muscle mass, (**D**) Arm muscle mass, (**E**) Trunk muscle mass, and (**F**) Leg muscle mass were significantly lower in the sarcopenia group than in the non-sarcopenia group. * *p* < 0.05, ** *p* < 0.01.

**Figure 4 cancers-15-03243-f004:**
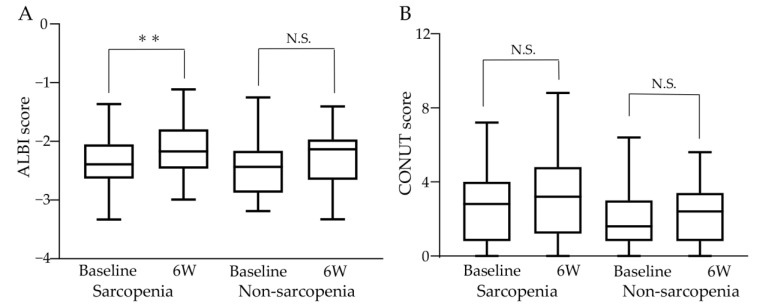
Changes in liver function and nutritional status during atezo/bev therapy. (**A**) In the sarcopenia group, the ALBI score increased significantly from baseline to after 6 weeks of treatment, and liver function worsened. In the non-sarcopenia group, the ALBI score did not change significantly before and after 6 weeks of treatment. (**B**) CONUT score did not change significantly between baseline and 6 weeks post-treatment in the sarcopenia and non-sarcopenia groups. ALBI: albumin-bilirubin, CONUT: controlling nutritional status, 6W: 6 weeks N.S.: not significant, ** *p* < 0.01.

**Figure 5 cancers-15-03243-f005:**
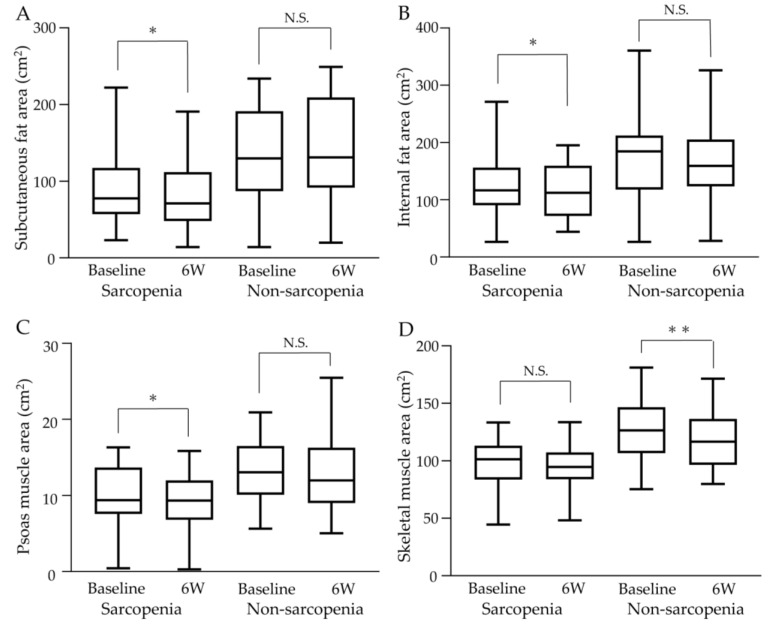
Changes in fat and muscle areas at the third lumbar vertebrae during atezo/bev therapy. (**A**) Subcutaneous fat area decreased significantly after treatment in the sarcopenia group, while the non-sarcopenia group showed no significant change after treatment. (**B**) Internal fat area decreased significantly after treatment in the sarcopenia group, while the non-sarcopenia group showed no significant change. (**C**) Psoas muscle area decreased significantly after treatment in the sarcopenia group, while the non-sarcopenia group showed no significant change. (**D**) Although skeletal muscle area showed no significant change before or after treatment in the sarcopenia group, a significant decrease was observed after treatment in the non-sarcopenia group. 6W: 6 weeks, N.S.: not significant, * *p* < 0.05, ** *p* < 0.01.

**Figure 6 cancers-15-03243-f006:**
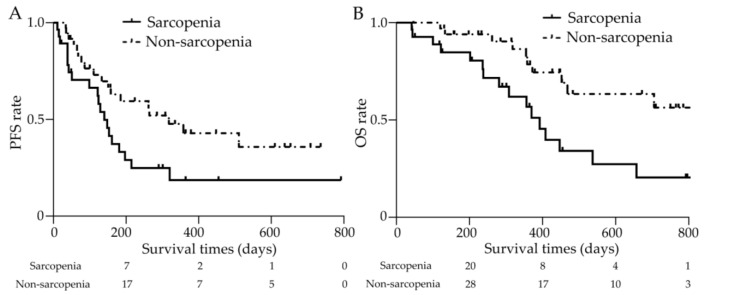
Clinical outcomes with or without sarcopenia. (**A**) The median PFS was 4.7 (0.4–26.4) and 10.6 (1.1–24.5) months in the sarcopenia and non-sarcopenia groups, respectively, with significant differences (*p* < 0.05). (**B**) The median OS was 12.6 (1.4–27.7) months in the sarcopenia group and was not achieved in the non-sarcopenia group. The Kaplan—Meier survival curves for both groups showed a significant difference (*p* < 0.01). PFS: progression-free survival, OS: overall survival.

**Table 1 cancers-15-03243-t001:** Baseline patient characteristics.

Characteristics	Sarcopenia Group	Non-Sarcopenia Group	*p* Value
	*n* = 28	*n* = 36	
Age (years): median (range)	79.5 (67–91)	71.5 (42–86)	<0.01
Sex (*n*): Male/Female (%)	21 (75.0)/7 (25.0)	28 (77.8)/8 (22.2)	>0.99
Height (m): median (range)	1.59 (1.36–1.70)	1.60 (1.42–1.77)	<0.05
Body weight (kg): median (range)	54.9 (42.8–71.6)	64.6 (46.4–84.0)	<0.01
BMI (kg/m^2^): median (range)	22.0 (18.3–28.3)	24.8 (19.2–37.6)	<0.01
GS (kg): median (range)	17.5 (6.5–27.1)	31.1 (15.0–38.4)	<0.01
BIA-SMI (kg/m^2^): median (range)	5.75 (4.26–6.98)	7.13 (5.40–10.55)	<0.01
Performance status (*n*): 0/1/2/3 (%)	10 (35.7)/8 (28.6)/9 (32.1)/1 (3.6)	27 (75.0)/7 (21.2)/2 (5.6)/0 (0.0)	<0.01
CONUT score (*n*): 0–1/2–4/5–8/>8 (%)	9 (32.1)/9 (32.1)/9 (32.1)/1 (3.6)	10 (27.8)/19 (52.8)/5 (13.9)/2 (5.6)	0.24
Etiology (*n*): HBV/HCV/ALD/NAFLD/PBC (%)	1 (3.6)/10 (35.7)/9 (32.1)/7 (25.0)/1 (3.6)	1 (2.8)/11 (30.6)/13 (36.1)/10 (27.8)/1 (2.8)	0.99
Previous treatment with MTAs (*n*): Yes/No (%)	11 (39.3)/17 (60.7)	10 (27.8)/26 (72.2)	0.42
Child-Pugh classification (*n*): A/B (%)	26 (92.9)/2 (7.1)	33 (91.7)/3 (8.3)	>0.99
mALBI grade (*n*): 1/2a/2b/3 (%)	8 (28.6)/9 (32.1)/10 (35.7)/1 (3.6)	15 (31.7)/7 (21.2)/13 (36.1)/1 (2.8)	0.62
AFP (ng/mL): median (range)	74 (2–628,992)	13 (2–106,190)	0.20
DCP (mAU/mL): median (range)	1433 (12–318,535)	344 (15–225,133)	0.31
Maximum tumor size (*n*): <5 cm/≥5 cm (%)	12 (42.9)/16 (57.1)	20 (55.6)/16 (44.4)	0.45
Number of tumors (*n*): ≤3/≥4 (%)	12 (42.9)/16 (57.1)	15 (41.7)/21 (58.3)	>0.99
Major vascular invasion (*n*): Yes/No (%)	9 (32.1)/19 (67.9)	6 (16.7)/30 (83.3)	0.24
Extrahepatic metastasis (*n*): Yes/No (%)	9 (32.1)/19 (67.9)	8 (21.6)/28 (77.8)	0.41
TMN staging LCSGJ 6th (*n*): III/IVa/IVb (%)	14 (50.0)/6 (21.4)/8 (28.6)	24 (66.7)/5 (13.9)/7 (19.4)	0.40
BCLC staging (*n*): B/C (%)	11 (39.3)/17 (60.7)	23 (63.9)/13 (36.1)	0.08

BMI: body mass index, GS: grip strength, BIA: bioelectrical impedance analysis, SMI: skeletal muscle index, CONUT: controlling nutritional status, HBV: hepatitis B virus, HCV: hepatitis C virus, ALD: alcoholic liver disease, NAFLD: non-alcoholic fatty liver disease, PBC: primary biliary cholangitis, MTAs: molecular target agents, mALBI: modified albumin-bilirubin, AFP: alpha-fetoprotein, DCP: des-γ-carboxy prothrombin, TNM: tumor-node-metastasis, LCSGJ: Liver Cancer Study Group of Japan, BCLC: Barcelona Clinic Liver Cancer.

**Table 2 cancers-15-03243-t002:** Therapeutic effect by modified RESIST after 6 weeks of atezo/bev treatment.

Therapeutic Effect	Sarcopenia Group	Non-Sarcopenia Group	*p* Value
	*n* = 28	*n* = 36	
ORR, *n* (%)	7 (25.0)	14 (38.9)	0.29
DCR, *n* (%)	21 (75.0)	29 (80.6)	0.76
CR, *n* (%)	1 (3.6)	0 (0.0)	
PR, *n* (%)	6 (21.4)	14 (38.9)	
SD, *n* (%)	14 (50.0)	15 (41.7)	
PD, *n* (%)	7 (25.0)	7 (19.4)	

Atezo: atezolizumab, Bev: bevacizumab, ORR: overall response rate, DCR: disease control rate, CR: complete response, PR: partial response, SD: stable disease, PD: progressive disease.

**Table 3 cancers-15-03243-t003:** AEs in HCC patients with atezo/bev therapy.

	Sarcopenia Group	Non-Sarcopenia Group	*p* Value
	*n* = 28	*n* = 36	
Any AEs, *n* (%)	25 (89.3)	24 (66.7)	<0.05
Grade 1	4 (14.3)	6 (16.7)	
Grade 2	6 (21.4)	10 (27.8)	
Grade 3	12 (42.9)	8 (22.2)	
Grade 4	3 (10.7)	0 (0.0)	
Grade 5	0 (0.0)	0 (0.0)	
Major AEs, *n* (%)			
Anorexia	18 (64.3)	7 (19.4)	<0.01
Proteinuria	9 (32.1)	10 (27.8)	0.79
Diarrhea	8 (28.6)	1 (2.8)	<0.01
Mucositis oral	5 (17.9)	4 (11.1)	0.49
Eczema	3 (10.7)	0 (0.0)	0.08
Nausea	3 (10.7)	1 (2.8)	0.31
Ascites	3 (10.7)	1 (2.8)	0.31
Hypertension	2 (7.1)	7 (19.4)	0.28
Severe AEs of grade ≥ 3, *n* (%)	14 (50.0)	7 (19.4)	<0.05
Anorexia	6 (21.4)	0 (0.0)	
Diarrhea	3 (10.7)	0 (0.0)	
Mucositis oral	3 (10.7)	1 (2.8)	
Ascites	2 (7.1)	1 (2.8)	
Proteinuria	1 (3.6)	1 (2.8)	
Tumor hemorrhage	1 (3.6)	0 (0.0)	
Pulmonary fibrosis	1 (3.6)	1 (2.8)	
Myocardial infarction	1 (3.6)	0 (0.0)	
Biliary tract infection	1 (3.6)	0 (0.0)	
Upper gastrointestinal hemorrhage	1 (3.6)	0 (0.0)	
Palmar-plantar erythrodysesthesia syndrome	1 (3.6)	0 (0.0)	
Tumor lysis syndrome	1 (3.6)	1 (2.8)	
Aspartate aminotransferase increased	0 (0.0)	1 (2.8)	
Platelet count decreased	0 (0.0)	1 (2.8)	
Acute kidney injury	0 (0.0)	1 (2.8)	
Pleuritic pain	0 (0.0)	1 (2.8)	

AEs: adverse events.

**Table 4 cancers-15-03243-t004:** Univariate and multivariate of OS.

Variable		Univariate Analysis			Multivariate Analysis		
		HR	95% CI	*p* Value	HR	96% CI	*p* Value
Sex	Male	0.73	0.32–1.79	0.46			
Age	≥80 years	2.63	1.15–5.86	<0.05	1.54	0.57–4.09	0.39
BMI	<23 kg/m^2^	1.86	0.81–4.17	0.13			
SMI	Male < 7.0 kg/m^2^ or Female < 5.7 kg/m^2^	1.92	0.84–4.96	0.14			
Sarcopenia	Yes	2.58	1.17–5.95	<0.05	2.97	1.22–7.66	<0.05
PS	≥1	3.30	1.48–7.64	<0.01	1.81	0.58–5.64	0.30
Etiology	ALD	1.47	0.64–3.25	0.34			
Previous treatment with MTAs	Yes	0.77	0.32–1.73	0.54			
Child-Pugh score	≥6	4.11	1.82–10.11	<0.01	4.18	1.51–12.30	<0.01
Maximum tumor size	≥50 mm	2.92	1.30–6.94	<0.05	2.11	0.88–5.36	0.10
Number of tumors	≥4	0.88	0.34–1.96	0.74			
Major vascular invasion	Yes	2.51	1.01–5.71	<0.05	0.50	0.15–1.55	0.24
Extrahepatic metastasis	Yes	1.40	0.59–3.11	0.42			

OS: overall survival, BMI: body mass index, SMI: skeletal muscle index, PS: performance status, ALD: alcoholic liver disease, MTAs: molecular target agents, CI: confidence interval.

## Data Availability

The data that support the findings of this study are not publicly available because they contain information that could compromise the privacy of research participants, but are available from the corresponding author [K.O.] upon reasonable request.

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
