# Peer review of "Relationship between Accurate Diagnosis of Sarcopenia and Prognosis in Patients with Hepatocellular Carcinoma Treated with Atezolizumab plus Bevacizumab Combination Therapy"

_cancers, 2023, doi:10.3390/cancers15123243_

Round 1

Reviewer 1 Report (Previous Reviewer 2)

The authors have addressed all my points and improved the manuscript substantially.

Overall, the use of the english language is sufficient.

Author Response

  1. The authors have addressed all my points and improved the manuscript substantially. Overall, the use of the english language is sufficient.

Response: Thank you very much for your comments.

In accordance with reviewer’s comments, we have revised and corrected our manuscript and provide the necessary additional information.

Reviewer 2 Report (Previous Reviewer 3)

I have no additional comments 

English language use, punctuation and syntax are good 

Author Response

  1. I have no additional comments. English language use, punctuation and syntax are good.

Response: Thank you very much for your comments.

In accordance with reviewer’s comments, we have revised and corrected our manuscript and provide the necessary additional information.

Reviewer 3 Report (New Reviewer)

Very interesting paper. I have only some minor comments:

1) Did sarcopenia influence also the safety profile of these drugs? In this regard, cite the recent MA assessing the risk of flare of pre-existing immuno-related disease in patients under ICIs therapy (PMID: 33314269)

2) The sample size is limited. This should be commented among the limitations to the paper

Author Response

1) Did sarcopenia influence also the safety profile of these drugs? In this regard, cite the recent MA assessing the risk of flare of pre-existing immuno-related disease in patients under ICIs therapy (PMID: 33314269).

Response: Thank you very much for your comments.

As shown in Table 3, 25 (89.3%) of the 28 patients in the sarcopenia group had AEs of any grade, while 24 (66.7%) of 36 patients in the non-sarcopenia group had AEs of any grade, indicating that the sarcopenia group had significantly more AEs than the non-sarcopenia group. In the sarcopenia group, anorexia was observed in 18 (64.3%) patients, proteinuria in nine (32.1%), diarrhea in eight (28.6%), mucositis in five (17.9%), eczema in three (10.7%), and nausea in three (10.7%). In the non-sarcopenia group, proteinuria was observed in 10 patients (27.8%), anorexia in seven (19.4%), hypertension in seven (19.4%), and oral mucositis in four (11.1%). Grade 3 or higher AEs were observed in 14 (50.0%) patients in the sarcopenia group and seven (19.4%) patients in the non-sarcopenia groups, respectively, indicating that the sarcopenia group had significantly more severe AEs than the non-sarcopenia group. The significantly increased incidence of AEs with atezo/bev therapy in patients with sarcopenia, which may be associated with worse clinical outcomes, suggests that careful management of AEs may contribute to better clinical outcomes when treating uHCC patients with sarcopenia.

Unresectable HCC patients on atezo/bev therapy should be observed more closely over a period shorter than 3 weeks, and AE should be carefully addressed. In addition, branched-chain ami-no acid formulation should be actively considered, as evidence has been estab-lished that prescribing BCAA contributes to the maintenance of liver function (Clin Gastroenterol Hepatol. 3(7), 705-713, 2005) (Gattroenterology. 124(7), 1792-1801, 2003). In fact, BCAA administration has been reported to be important not only for improving liver function but also for maintaining skeletal muscle mass (J Gastroenterol. 53(3), 427-437, 2018), which may have a dual bene-fit for patients with sarcopenia. Another report suggests that L-carnitine administration are useful in preventing skeletal muscle loss in patients with cirrhosis (Am J Clin Nutr. 93(4), 799-808, 2011), and may be an important strategy in the multidisciplinary treatment for uHCC. Furthermore, a previous systematic review suggested that ICIs caused more AEs and affected clinical outcomes in patients with a history of autoimmune disease (Aliment Pharmacol Ther. 53(3), 374-382, 2021), and sarcopenic patients should be evaluated more carefully for preexisting conditions, including autoimmune disease.

We agree with your opinion and  have revised the corrsponding parts of the result and discussion section.

2) The sample size is limited. This should be commented among the limitations to the paper.

Response: As you pointed out, it was a retrospective study, and the number of patients included was insufficient because grip strength measurement and body composition examination by bioelectrical impredance analysis were often not routinely performed before atezo/bev treatment. Therefore, a study with a larger number of patients is necessary.

We have revised the corresponding parts of the result section added this point as a limitation in the discussion section accordinng to your suggestion.

Thank you for your kind comment.

This manuscript is a resubmission of an earlier submission. The following is a list of the peer review reports and author responses from that submission.

Round 1

Reviewer 1 Report

The article is well written, and all results are well presented and discussed. Limitations of this study also been highlighted in the discussion. 

Reviewer 2 Report

In their manuscript Oura et al. report on a retrospective study investigating the prognostic value of sarcopenia diagnosed by grip strength (GS) and bioelectrical impedance analysis (BIA)-skeletal muscle index (SMI) in patients with unresectable hepatocellular carcinoma (uHCC) treated with atezolizumab/bevacizumab.

The retrospective multi-center study conducted in Japan included 58 uHCC patients on atezo/bev therapy. The patients were allocated to the sarcopenia group if the pretreatment GS or BMI-SMI were below the cut-off values (GS: male <28kg, female <18kg; BIA-SMI: male <8.0 kg/m2, female <5.7 kg/m2), or the non-sarcopenia group if the GS or BMI-SMI were above the cut-off values. The authors report that sarcopenia was significantly associated with poor clinical outcomes, including more frequent adverse events and decreased liver function, and reduced median overall survival compared to non-sarcopenia patients.

While this is the first study to investigates sarcopenia measured by GS and SMI in patients with uHCC receiving immunotherapy, the association of sarcopenia with worse outcome measures in HCC has been reported previously for sorafenib (Takada H et al.), lenvatinib (Uojima H et al.) and atezolizumab/bevacizumab (Matsumoto H et al.). Thus, the novelty and relevance of these findings for the field are limited.

Major points:

1.     In general, the sample size for each cohort was relatively small making statistically meaningful comparisons difficult.

2.     The sarcopenia group was significantly older than the non-sarcopenia cohort. No information was given on comorbidities, which could influence outcome by itself.

3.     Performance status in the sarcopenia group was significantly worse than in the non-sarcopenia cohort, which again could be a bias for worse outcome.

4.     No information on causes of death were provided. Was the reduced OS due to HCC-related factors?

Minor points:

1.     How was anorexia defined in the AEs table? Since the differences in AEs between the two groups are mostly attributable to a higher prevalence of anorexia in the sarcopenia cohort, this should be explained in more detail.

2.     The impact of the findings of the study on clinical practice should be discussed in more detail. How would the authors approach patients with sarcopenia and uHCC differently?  

Reviewer 3 Report

Thank you for your sudy

A few suggestions.

Simple Summary – The first sentence doesn't really make sense – the initial comment that sarcopenia related factors affect efficacy in systemic therapies for HCC is true but then the authors comment that there is considerably less evidence where Atezo/Bev are used – which is itself a systemic therapy. The sentence should be reworded.

- Page 2 Line 72 – The Atezo/Bev combination regimen ………………….is currently the most effective treatment strategy for HCC as first-line treatment. Although a reference is given in the subsequent sentence it is in referral to the standard of care Sorafanib which is arguable no longer the case. Atezo/Bev is certainly a choice but there is not enough evidence to indicate it is the most effective treatment strategy out of the current options – a reasonable comment would be that – Since the results of the IMbrave150 study Atezo/Bev is considered the most effective first line combination therapy ---.

There was a significant difference between the mean ages of the two groups. Could this difference alone have been responsible for the differences observed. This age difference should perhaps be put in a a limitation of this study and future studies should attempt to have matched age groups. The relationship between sarcopenia and age could also be elaborated on.

The discussion section is a bit lengthy but well written.